# Development of Glycosylation-Modified ^D^PPA-1 Compounds as Innovative PD-1/PD-L1 Blockers: Design, Synthesis, and Biological Evaluation

**DOI:** 10.3390/molecules29081898

**Published:** 2024-04-22

**Authors:** Peng Deng, Xiaodan Dong, Ziyuan Wu, Xixi Hou, Longfei Mao, Jingjing Guo, Wenshan Zhao, Chune Peng, Zhe Zhang, Lizeng Peng

**Affiliations:** 1Key Laboratory of Novel Food Resources Processing Ministry of Agriculture, Key Laboratory of Agro-Products Processing Technology of Shandong Province, Institute of Agro-Food Science and Technology, Shandong Academy of Agricultural Sciences, Jinan 250100, China; 2College of Basic Medicine and Forensic Medicine, Henan University of Science and Technology, 263 Kaiyuan Road, Luoyang 471003, China; 3School of Chemistry and Chemical Engineering, Henan Normal University, Xinxiang 453007, China; 4Centre for Artificial Intelligence Driven Drug Discovery, Faculty of Applied Sciences, Macao Polytechnic University, Macao 999078, China; jguo@mpu.edu.mo; 5School of Life Sciences, Zhengzhou University, Zhengzhou 450001, China; zhaowsh07@zzu.edu.cn; 6School of Sciences, Henan University of Technology, Zhengzhou 450001, China

**Keywords:** cancer immunotherapy, immune checkpoint, PD-1/PD-L1, peptide, glycosylation

## Abstract

In the context of peptide drug development, glycosylation plays a pivotal role. Accordingly, L-type peptides were synthesized predicated upon the PD-1/PD-L1 blocker ^D^PPA-1. Subsequent glycosylation resulted in the production of two distinct glycopeptides, D-glu-^L^PPA-1 and D-gal-^L^PPA-1, by using D-glucose (D-glu) and D-galactose (D-gal), respectively, during glycosylation. Both glycopeptides significantly inhibited the interaction between PD-1 and PD-L1, and the measured half maximal inhibitory concentrations (IC_50s_) were 75.5 μM and 101.9 μM for D-glu-LPPA-1 and D-gal-LPPA-1, respectively. Furthermore, D-gal-LPPA-1 displayed a pronounced ability to restore T-cell functionality. In an MC38 tumor-bearing mouse model, D-gal-LPPA-1 demonstrated a significant inhibitory effect. Notably, D-gal-LPPA-1 substantially augmented the abundance and functionality of CD8^+^ T cells in the tumor microenvironment. Additionally, in the lymph nodes and spleens, D-gal-LPPA-1 significantly increased the proportion of CD8^+^ T cells secreting interferon-gamma (IFN-γ). These strong findings position D-gal-LPPA-1 as a potent enhancer of the antitumor immune response in MC38 tumor-bearing mice, underscoring its potential as a formidable PD-1/PD-L1 blocking agent.

## 1. Introduction

Glycosylation is a posttranslational modification within the proteomic landscape of living organisms and constitutes a foundational aspect of biological processes [1,2]. Approximately half of the identified proteins are characterized as glycoproteins, revealing a spectrum of types, properties, and functions [3,4]. This diversity is underscored by the pivotal roles played by glycoproteins in an array of physiological and pathological phenomena, including cell adhesion and recognition, tissue differentiation and development, immune regulation, tumor metastasis, inflammation, and pathogen infection [1,5]. The intricate sugar side chains of glycoproteins can modulate the conformation and physicochemical properties of proteins, thereby exerting profound influences on their biological activities [6]. The consequential importance of these activities has garnered substantial interest in the exploration of glycoproteins.

However, the inherent microheterogeneity of glycoproteins is characterized by the presence of various glycosylation products within a given protein sequence. However, because of their limited abundance in living organisms, isolating individual glycoproteins for detailed structural characterization and activity studies is a formidable challenge [7]. In contrast, glycopeptides, while exhibiting less structural complexity and lower molecular weights than glycoproteins, effectively preserve the crucial carbohydrate–peptide linkage region of glycoprotein cores [8,9]. Serving as invaluable models for the investigation of glycoproteins, glycopeptides also play pivotal roles in diverse physiological and pathological functions within animals and plants, thereby elevating their prominence as a focal point in peptide drug development research [10,11,12]. Beyond their utility as research models, glycosylated peptides offer a compelling solution to the inherent weaknesses plaguing conventional peptides, such as their proclivity toward poor chemical and physical stability. These glycosylated counterparts effectively address concerns associated with the abbreviated half-life, instability, and susceptibility to degradation within the body, commonly observed in ordinary peptides [13].

PD-1, a cell surface receptor expressed widely on immune cells such as, B cells, T cells, regulatory T cells (Tregs), dendritic cells (DCs), natural killer cells (NKs), and macrophages plays a crucial role in immune regulation [14,15,16,17]. Functioning as an intrinsic negative regulator, PD-1 is involved in dampening antigen-specific T-cell responses, particularly in conditions such as viral infections and cancer [18]. The primary ligand of PD-1 is PD-L1, which is expressed on various immune-activating tissues or cells, such as bone marrow-derived mast cells, dendritic cells, mesenchymal stem cells (MSCs), monocytes, T lymphocytes, B lymphocytes, and various immune-privileged organs under normal physiological circumstances [17,19]. Studies have revealed that PD-L1 upregulation on tumor cells induces the expression of interferon-gamma (IFN-γ) [20]. The overexpression of PD-L1 on tumor cells strategically aids in evading immune cell surveillance. The binding of PD-L1 to the PD-1 receptor results in the inactivation of tumor-infiltrating lymphocytes (TILs), leading to subsequent apoptosis of tumor-specific T cells [21]. Moreover, PD-L1 could be a selective therapeutic target for cancer treatment because of the low expression of PD-L1 in normal human tissues.

Despite the success of monoclonal antibodies targeting the PD-1/PD-L1 signaling pathway, exemplified by FDA-approved drugs such as nivolumab, pembrolizumab, and tislelizumab [22], certain limitations persist, such as high immunogenicity, high production costs, and restricted tumor tissue penetration. In addressing these challenges, an increasing number of researchers are investigating the role of peptide drugs in the PD-1/PD-L1 signaling pathway [23,24,25,26]. Peptide drugs possess distinct advantages over antibody drugs, such as the feasibility of oral administration, reduced manufacturing costs, and improved penetration into tumor tissues [27,28]. An illustrative example is ^D^PPA-1, a peptide drug designed to selectively target PD-L1, disrupting the interaction between PD-1 and PD-L1. In a CT26 xenograft mouse model, ^D^PPA-1 demonstrated significant efficacy in inhibiting tumor growth [29]. Another noteworthy contender macrocyclic peptide BMS-986189, developed by Bristol Myers Squibb (Tokyo, Japan) is currently undergoing clinical trials [30]. This peptide further underscores the promising potential of PD-1/PD-L1-targeted peptide drugs in the realm of antitumor therapy.

Peptide drugs play an important role in the development of PD-1/PD-L1 inhibitors. Therefore, this study focused on ^D^PPA-1. An analogous L-type peptide, denoted ^L^PPA-1, was subjected to glycosylations using D-glucose (D-glu) and D-galactose (D-gal), yielding two glycopeptides, namely, D-glu-^L^PPA-1 and D-gal-^L^PPA-1. Subsequent functional investigations were carried out to assess the potential therapeutic effects of these compounds.

## 2. Results and Discussion

### 2.1. The Ability of D-glu-^L^PPA-1 and D-gal-^L^PPA-1 to Block PD-1/PD-L1 Interaction

The interaction between PD-1 and PD-L1 plays a pivotal role in suppressing T-cell function [31], and disrupting this interaction has emerged as a viable therapeutic strategy in cancer treatment. Thus, we investigated the inhibitory potential of D-glu-^L^PPA-1 and D-gal-^L^PPA-1 on the PD-1/PD-L1 interaction. To conduct the experiment, CHO-K1-hPD-1 cells were harvested at a density of 3 × 10^5^ cells per sample. Peptides were solubilized to a concentration of 200 μM and serially diluted before coincubation with 50 ng of the hPD-L1 protein for 30 min. The protein–peptide mixture was introduced into CHO-K1-hPD-1 cells, which were then coincubated for an additional 30 min. The anti-Fc PE antibody was then added and incubated for 30 min. Flow cytometry was used to evaluate the efficacy of the peptides in blocking the hPD-1/hPD-L1 interaction. The results from the blocking assay revealed that both D-glu-^L^PPA-1 and D-gal-^L^PPA-1 effectively impeded the binding of PD-1 to PD-L1, with calculated IC_50_ values of 75 μM (D-glu-^L^PPA-1) and 101 μM (D-gal-^L^PPA-1) (Figure 1).

### 2.2. Molecular Docking Studies of D-glu-^L^PPA-1 and D-gal-^L^PPA-1 to PD-1/PD-L1

In order to intuitively analyze the binding mode of the designed glycopeptides to PD-1/PD-L1, we selected D-glu-^L^PPA-1 and D-gal-^L^PPA-1 as model compounds (Figure 1). The results indicate that D-glu-^L^PPA-1 can interact with Asp26, Leu27, Asp122, Tyr123, Lys124, and Arg125, among which Asp122, Tyr123, Lys124, and Arg125 are crucial binding sites of PD-L1 protein with PD-1 protein (Figure 2A). D-gal-^L^PPA-1 can interact with Gln66, Tyr123, Arg113, and Asp61, among which Glu58, Gln66, and Tyr123 are crucial binding sites of PD-L1 protein with PD-1 protein. Through molecular docking, we can intuitively observe that the D-glu structure in D-glu-^L^PPA-1 can form hydrogen bonds with Asp122, while the D-gal structure in D-gal-^L^PPA-1 can interact via hydrogen bonding with Gln66. Therefore, we further demonstrate that D-glu-^L^PPA-1 and D-gal-^L^PPA-1 have the ability to block the binding of PD-L1 protein and PD-1 protein (Figure 2B).

### 2.3. D-gal-^L^PPA-1 Enhanced IL-2 Secretion in a Jurkat Cell Coculture Assay

We further investigated whether the peptides D-glu-^L^PPA-1 and D-gal-^L^PPA-1 could enhance the function of T cells. The peptides were cocultured with Jurkat (PHA and PMA prestimulated) and CHO-K1-hPD-L1 cells, and OPBP-1 was used as a positive control. At a concentration of 100 μM, the peptide D-gal-^L^PPA-1 stimulated the secretion of IL-2 by CD45+ Jurkat T cells, which was greater than that stimulated by OPBP-1 (Figure 3).

### 2.4. D-gal-^L^PPA-1 Exhibited High Binding Affinity for PD-L1

Subsequently, microscale thermophoresis (MST) experiments were conducted to assess the affinity of D-gal-^L^PPA-1 for the PD-L1 protein. The MST results demonstrated that the dissociation constant (K_D_) values of D-gal-^L^PPA-1 for mouse PD-L1 (mPD-L1) proteins and human PD-L1 (hPD-L1) proteins were 0.0149 µM and 0.0470 µM, respectively (Figure 4).

### 2.5. D-gal-^L^PPA-1 Significantly Inhibited MC38 Tumor Growth In Vivo

An MC38 tumor-bearing mouse model was used to explore the antitumor effects of the peptide D-gal-^L^PPA-1. The results demonstrated that intraperitoneal injection of the peptide at doses of 1 mg/kg and 3 mg/kg significantly inhibited tumor volume and tumor weight compared to those in the control group, with the 3 mg/kg dose exhibiting a more potent antitumor effect. Additionally, no significant changes in body weight were observed during the administration of the peptide (Figure 5).

### 2.6. The Toxicity of D-gal-^L^PPA-1 In Vivo

Additionally, major organs were subjected to H&E staining. As shown in Figure 6, after drug treatment, there was no apparent toxicity observed in the organs such as the heart, liver, spleen, lung, or kidney, compared to that in the control group. These results illustrate the safety of D-gal-^L^PPA-1 for cancer treatment.

### 2.7. D-gal-^L^PPA-1 Significantly Enhances the Functionality of CD8^+^ T Cells

The in vitro experiments were also conducted to further analyze the antitumor mechanism of the peptide. The results revealed that D-gal-^L^PPA-1 significantly enhanced the quantity of CD8^+^ T cells at the tumor site, as well as the proportion of CD8^+^ T cells secreting IFN-γ at the tumor site. Furthermore, D-gal-^L^PPA-1 significantly increased the proportion of IFN-γ-secreting CD8^+^ T cells in the lymph nodes and spleen. These findings suggested that D-gal-^L^PPA-1 effectively enhanced the antitumor immune response in MC38 tumor-bearing mice (Figure 7).

The D-glu-^L^PPA-1 and D-gal-^L^PPA-1 in this study exhibited robust inhibitory activity against the interaction between PD-1 and PD-L1. In addition, D-gal-^L^PPA-1 significantly improved the T-cell functionality and binding affinity. Through molecular docking, we can intuitively observe that the D-glu structure in D-glu-^L^PPA-1 can form hydrogen bonds with Asp122 of PD-1/PD-L1, while the D-gal structure in D-gal-^L^PPA-1 can interact via hydrogen bonding with Gln66 of PD-1/PD-L1. Hence, the glycosylation of peptides through D-glu and D-gal has emerged as a viable strategy for the development of PD-1/PD-L1 inhibitors.

## 3. Materials and Methods

### 3.1. Solid-Phase Peptide Synthesis

The L-type peptide Asn-Tyr-Ser-Lys-Pro-Thr-Asp-Arg-Gln-Tyr-His-Phe (^L^PPA-1) was synthesized by Fmoc solid-phase peptide synthesis. Subsequently, glycosylations were independently performed to produce two glycopeptides: D-glu-^L^PPA-1, which was glycosylated with D-glucose, and D-gal-^L^PPA-1, which was glycosylated with D-galactose. Structural confirmation of the target compounds was performed through mass spectrometry (MS). The two glycopeptides were synthesized in-BGI.

#### 3.1.1. Preparation of Fmoc-Phe-CTC Resin



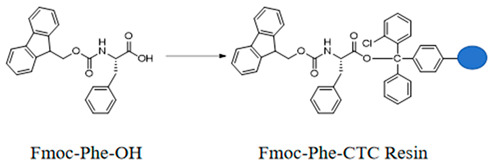



Weigh 0.3 g of CTC Resin with a loading capacity of 1.0 mmol/g and 0.4 mmol of Fmoc-Phe-OH into a peptide synthesis reactor, add 10 mL of DCM and 0.9 mmol of DIPEA, react for 3 h, then add 1 mL of methanol, stir the reaction for 30 min, filter, and sequentially wash the filter cake twice with DMF, twice with methanol, and twice with DCM to obtain Fmoc-Phe-CTC Resin.

#### 3.1.2. Preparation of Phe-CTC Resin



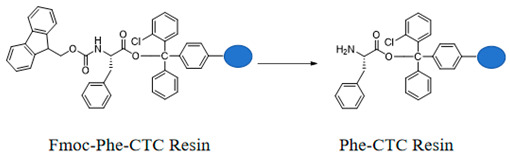



Add the Fmoc-Phe-CTC Resin to *N*,*N*-dimethylformamide containing 20% piperidine. Stir the reaction under nitrogen protection for 60 min. After filtration, wash the resin five times with *N*,*N*-dimethylformamide to obtain Phe-CTC Resin.

#### 3.1.3. Preparation of Fmoc-His-Phe-CTC Resin



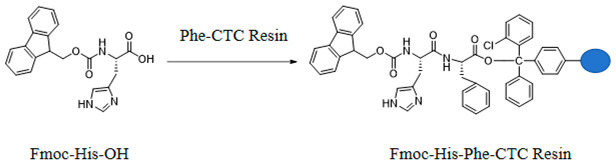



Add Fmoc-His-OH and Phe-CTC Resin to *N*,*N*-dimethylformamide, then sequentially add DIPEA and HBTU. Stir the reaction at room temperature for 60 min, then filter. Wash the filter cake three times with *N*,*N*-dimethylformamide to obtain Fmoc-His-Phe-CTC Resin.

#### 3.1.4. Preparation of D-glu-Asn-Tyr-Ser-Lys-Pro-Thr-Asp-Arg-Gln-Tyr-His-Phe-CTC Resin

Following the methods described in Section 3.1.1, Section 3.1.2 and Section 3.1.3, sequentially couple Tyr, Gln, Arg, Asp, Thr, Pro, Lys, Ser, Tyr, and D-glu-Asn onto the resin peptide chain to obtain D-glu-Asn-Tyr-Ser-Lys-Pro-Thr-Asp-Arg-Gln-Tyr-His-Phe-CTC Resin.

#### 3.1.5. Preparation of D-gal-Asn-Tyr-Ser-Lys-Pro-Thr-Asp-Arg-Gln-Tyr-His-Phe-CTC Resin

Following the methods described in Section 3.1.1, Section 3.1.2 and Section 3.1.3, sequentially couple Tyr, Gln, Arg, Asp, Thr, Pro, Lys, Ser, Tyr, and D-gal-Asn onto the resin peptide chain to obtain D-gal-Asn-Tyr-Ser-Lys-Pro-Thr-Asp-Arg-Gln-Tyr-His-Phe-CTC Resin.

#### 3.1.6. Preparation of D-glu-Asn-Tyr-Ser-Lys-Pro-Thr-Asp-Arg-Gln-Tyr-His-Phe (D-glu-^L^PPA-1, Figure 8A)

Add D-glu-Asn-Tyr-Ser-Lys-Pro-Thr-Asp-Arg-Gln-Tyr-His-Phe-CTC Resin to a mixture of trifluoroacetic acid, triisopropylsilane, water, and dithiothreitol (in a volume ratio of 90:5:2.5:2.5). Stir the reaction for 8 h, then add ether and continue stirring for 1 h. After filtration, dissolve the peptide in a mixture of acetonitrile and water (in a volume ratio of 1:1). Purify the peptide by reverse-phase high-performance liquid chromatography (HPLC) to obtain D-glu-LPPA-1. The purity detected by HPLC is 98.29%. The results of mass spectrometry (MS) are as follows: [M + 4H]^4+^: M = 430.5; [M + 3H]^3+^: M = 573.7; [M + 2H]^2+^: M = 859.8.

#### 3.1.7. Preparation of D-gal-Asn-Tyr-Ser-Lys-Pro-Thr-Asp-Arg-Gln-Tyr-His-Phe (D-gal-^L^PPA-1, Figure 8B)

Add D-gal-Asn-Tyr-Ser-Lys-Pro-Thr-Asp-Arg-Gln-Tyr-His-Phe-CTC Resin to a mixture of trifluoroacetic acid, triisopropylsilane, water, and dithiothreitol (in a volume ratio of 90:5:2.5:2.5). Stir the reaction for 8 h, then add ether and continue stirring for 1 h. After filtration, dissolve the peptide in a mixture of acetonitrile and water (in a volume ratio of 1:1). Purify the peptide by reverse-phase high-performance liquid chromatography (HPLC) to obtain D-gal-LPPA-1. The purity detected by HPLC is 100%. The results of mass spectrometry (MS) are as follows: [M + 4H]^4+^: M = 430.5; [M + 3H]^3+^: M = 573.6; [M + 2H]^2+^: M = 859.7.

**Figure 8 molecules-29-01898-f008:**
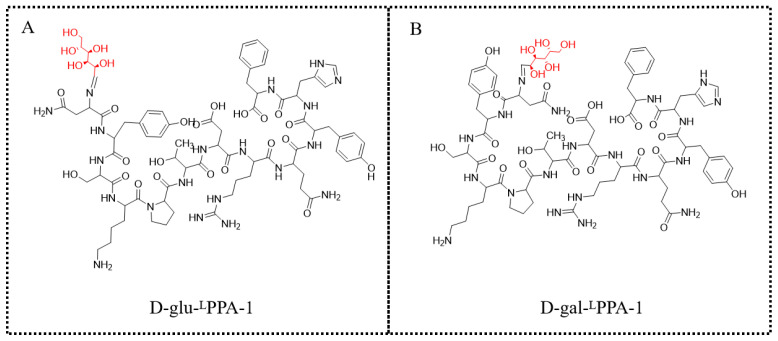
The structures of D-glu-^L^PPA-1 and D-gal-^L^PPA-1. (**A**) The structure of D-glu-^L^PPA-1. (**B**) The structure of D-gal-^L^PPA-1.

### 3.2. Chemicals and Reagents

Dimethylsulfoxide (DMSO) was obtained from Sigma-Aldrich (St. Louis, MI, USA). Dulbecco’s modified Eagle medium (DMEM), RPMI 1640 medium, fetal bovine serum (FBS), and penicillin/streptomycin were purchased from Gibco (Grand Island, NY, USA). An Enhanced Cell Counting Kit-8 (CCK-8), a Calcein/PI Live/Dead Viability Assay Kit, Giemsa dye and Reactive Oxygen Species (ROS) Assay Kit were obtained from Beyotime Biotechnology (Shanghai, China). An Annexin V-FITC/Propidium iodide (PI) staining kit and Matrigel matrix were obtained from BD Biosciences (Franklin Lake, NJ, USA). All of the other chemicals used were of analytical grade.

### 3.3. Cell Culture

The mouse colon carcinoma cell line MC38, the Chinese hamster ovary cell line CHO-K1-hPD-L1, the CHO-K1-mPD-L1 line, and the human T lymphocytic leukemia Jurkat cell line were maintained in RMPI 1640 (Corning, NY, USA). All the cells were cultured in medium supplemented with 10% FBS (Biological Industries, Cromwell, CT, USA) and 100 U/mL penicillin/streptomycin (Solarbio, Beijing, China) in an incubator with 5% CO_2_ at 37 °C.

We conducted experiments employing CHO-K1-hPD-L1 and CHO-K1-mPD-L1 cells previously constructed in the laboratory. The flow cytometry results indicated high expression of PD-L1 in both CHO-K1-hPD-L1 and CHO-K1-mPD-L1 cells. Subsequently, we employed flow cytometry to assess the expression of PD-L1 in MC38 cells. The results revealed a high level of PD-L1 expression in MC38 cells. Also, the heightened PD-L1 expression in MC38 cells has been substantiated in multiple studies [32,33,34]. Notably, our experimental paradigm involved the stimulation of Jurkat cells with PHA and PMA, a methodology documented in the literature, known to elicit PD-1 overexpression within Jurkat cells [35].

### 3.4. Peptide Blocking Assay

In the preliminary investigation, CHO-K1-hPD-1 cells were subjected to co-incubation with varying quantities of hPD-L1-Fc, specifically 200 ng, 100 ng, 50 ng, 25 ng, and 12.5 ng. Analysis of fluorescence intensities via flow cytometry revealed that the mean fluorescence intensity (MFI) value associated with the binding of 50 ng of hPD-L1-Fc to CHO-K1-hPD-1 cells approached approximately 1000, indicative of optimal binding efficiency. For subsequent multi-concentration blocking assays, peptides were prepared in concentration gradients spanning 200 μM, 100 μM, 50 μM, 25 μM, 12.5 μM, and 6.25 μM. Peptides at different concentrations were incubated with hPD-L1 (mPD-L1) protein (50 ng) on ice for 30 min. Subsequently, the mixture was added to CHO-K1-hPD-1 or CHO-K1-mPD-1 cells and coincubated on ice for 30 min. Subsequently, an anti-Fc PE antibody was added, and the mixture was incubated on ice for 30 min. The ability of the peptide to block the PD-1/PD-L1 interaction was assessed using flow cytometry [36].

### 3.5. Binding Assay

The concentrations of hPD-L1-His and mPD-L1-His proteins were adjusted to 200 nM and 800 nM, respectively, utilizing PBST as the diluent. Simultaneously, the concentration of RED-NHS676 dye was diluted to 100 nM in PBST. Subsequently, protein and dye were subjected to co-incubation at a volumetric ratio of 1:1 for a duration of 30 min, thereby facilitating the generation of labeled proteins. To establish peptide gradients, a progressive dilution was initiated from an initial concentration of 400 μM, eventually reaching 0.012 μM. Following this preparation, a homogenous amalgamation comprising 5 μL of labeled protein and 5 μL of small molecule diluent was extracted for further experimentation [36]. Subsequently, the samples were transferred to standard capillary tubes and analyzed by an MST instrument (Nano Temper, Monolith NT.115, München, Germany).

### 3.6. Coculture Assay

In the coculture experiment, Jurkat cells were stimulated with 12-myristic ester-13-acetate (PMA, 25 ng/mL) and phytohemagglutinin (PHA, 1 μg/mL) and cocultured with CHO-K1-hPD-L1 cells. Then, the cells were treated with 100 μM D-gal-^L^PPA-1 for 4 h. Then, 1 μL of protein transport inhibitor was added, and the coculture was incubated for 48 h. The cells were subsequently collected and incubated on ice for 30 min upon the addition of an anti-human CD45 FITC antibody. The cells were then fixed with 4% paraformaldehyde and incubated with anti-human-IL-2-APC (MQ1-17H12; eBioscience, San Diego, CA, USA) for 30 min. Flow cytometry was used for detection and analysis [37].

### 3.7. In Vivo Antitumor Experiment

Animal experiments were approved by the Zhengzhou University Ethics Committee. MC38 cells (1 × 10^6^) were subcutaneously inoculated into C57BL/N mice. The MC38 tumor-bearing mice were randomly allocated to three groupsn (*n* = 5 per group): a control group (NS, normal saline containing 1% DMSO); a D-gal-^L^PPA-1 peptide group (1 mg/kg); and a D-gal-^L^PPA-1 peptide group (3 mg/kg). Daily intraperitoneal injections of the peptide were initiated when the tumor volume reached 50–90 mm^3^. The tumor dimensions, including length, width, and height, were measured, and tumor volume was calculated using the following formula:V = 1/2 × A (length) × B (width) × C (height).

One day after the completion of administration, the mice were humanely sacrificed, and the tumor tissue, lymph nodes, and spleen were collected for subsequent experimental procedures [38].

Tumor cells, spleen cells, and lymph node cells were isolated from the tumor-bearing mice, and a single-cell suspension was prepared. The cells were plated and stimulated with 20 ng/mL 12-myristic ester-13-acetate (PMA; Sigma, Livonia, MI, USA) and 1 μM ionomycin (Sigma). During the plating process, protein transport inhibitors were added. After 4 h, the cells were collected for antibody staining. The relevant antibodies used included anti-mouse CD3 PerCP-eFluor710 (17A2, eBioscience), anti-mouse CD8α eFluor450 (53–6.7, eBioscience), and anti-CD45-FITC (30-F11). Following antibody incubation, staining was carried out using the intracellular marker anti-mouse IFN-γ APC (XMG1.2, eBioscience) or isotype control antibodies.

### 3.8. H&E Staining Analysis

Tumor tissue and heart, liver, spleen, lung, and kidney tissues were isolated from MC38 tumor-bearing mice and fixed in 4.0% paraformaldehyde. The samples were subsequently embedded in paraffin and subjected to hematoxylin and eosin staining. The results were analyzed using ImageJ1 software (NIH, Bethesda, MD, USA).

### 3.9. Statistical Analyses

The data were statistically analyzed using Graph Prism 7.0. A two-tailed Student’s *t* test or one-way analysis of variance followed by the Student–Newman–Keuls (SNK) test was used to assess significant differences. A *p* value < 0.05 was considered to indicate statistical significance. * *p* < 0.05, ** *p* < 0.01, and *** *p* < 0.001 based on the SNK test.

## 4. Conclusions

Based on the structure of the PD-1/PD-L1 peptide inhibitor ^D^PPA-1, the present study introduced the novel L-type peptide ^L^PPA-1. Subsequent glycosylations with D-glucose (D-glu) and D-galactose (D-gal) resulted in two glycopeptides: D-glu-^L^PPA-1 and D-gal-^L^PPA-1. The inhibitory efficacy of these compounds against PD-1 and PD-L1 was evaluated, revealing IC_50_ values of 75.5 μM and 101.9 μM for D-glu-^L^PPA-1 and D-gal-^L^PPA-1, respectively. Notably, D-gal-^L^PPA-1 exhibited significant potency in restoring T-cell functionality. In a murine model of MC38 tumors, D-gal-^L^PPA-1 demonstrated substantial suppression of tumor progression. Moreover, D-gal-^L^PPA-1 elicited a pronounced increase in the population of CD8^+^ T cells within the local tumor environment, coupled with an increase in the proportion of CD8^+^ T cells secreting IFN-γ. Furthermore, D-gal-^L^PPA-1 markedly elevated the percentage of CD8^+^ T cells secreting IFN-γ in the lymph nodes and spleen. These results collectively indicated that D-gal-^L^PPA-1 profoundly enhanced the antitumor immune response in MC38 tumor-bearing mice.

## Figures and Tables

**Figure 1 molecules-29-01898-f001:**
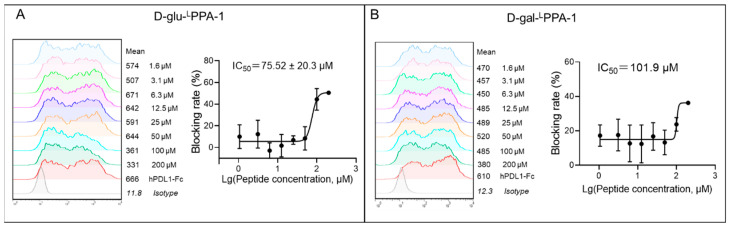
The ability of D-glu-^L^PPA-1 and D-gal-^L^PPA-1 to block the PD-1/PD-L1 interaction. (**A**) Flow cytometry results and the dose-dependent curve of D-glu-^L^PPA-1 in blocking the PD-1/PD-L1 interaction determined by flow cytometry from 200 μM to 1.6 μM. (**B**) Flow cytometry results and the dose-dependent curve of D-gal-^L^PPA-1 in blocking the PD-1/PD-L1 interaction determined by flow cytometry from 200 μM to 1.6 μM. The data are shown as the means ± SEMs from three independent biological triplicates.

**Figure 2 molecules-29-01898-f002:**
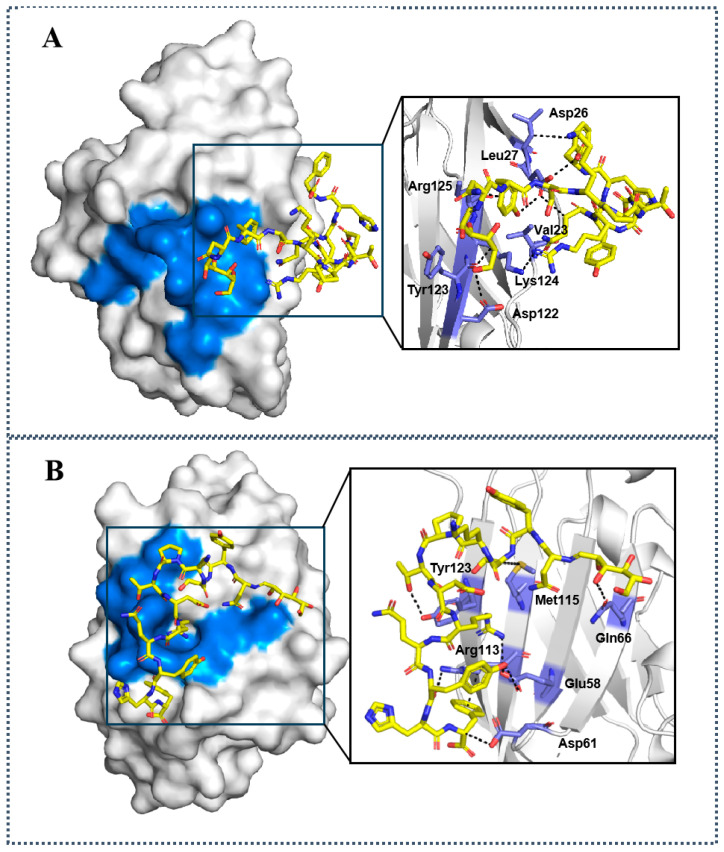
The binding modes of D-glu-^L^PPA-1 and D-gal-^L^PPA-1. (**A**) The binding modes of D-glu-^L^PPA-1. (**B**) The binding modes of D-gal-^L^PPA-1. The residues of hPD-1/hPD-L1 interaction interface were shown in blue. The peptides targeting PD-L1 were showed in yellow.

**Figure 3 molecules-29-01898-f003:**
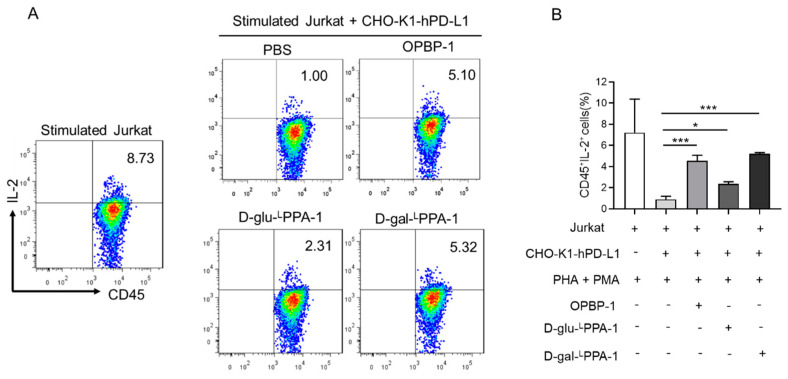
Flow cytometry and pooled data on IL-2 secretion in coculture experiments of Jurkat cells stimulated with PHA or PMA and CHO-K1-hPD-L1 cells. (**A**) Representative flow cytometry of IL-2 secretion by Jurkat cells. (**B**) Statistical diagram of IL-2 secretion by Jurkat cells. The bars in the graph depict the means ± SEMs derived from three distinct biological replicates. Significance levels are denoted as * *p* < 0.05 and *** *p* < 0.001 based on the SNK test.

**Figure 4 molecules-29-01898-f004:**
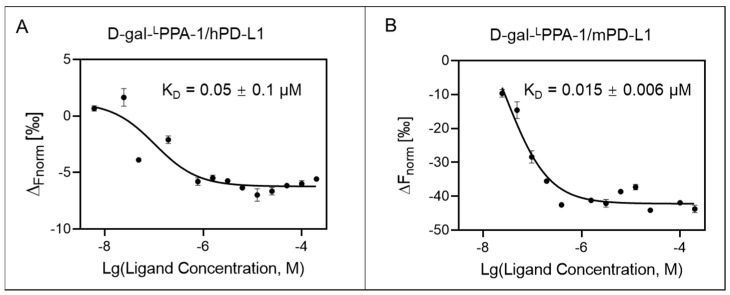
Dose-dependent binding curve of D-gal-^L^PPA-1 with PD-L1 determined by MST. Similar results were observed in triplicate. (**A**) The concentration-response curve of D-gal-^L^PPA-1 with hPD-L1. (**B**) The concentration-response curve of D-gal-^L^PPA-1 with mPD-L1. The data are presented as the means ± SEMs. The bars indicate the standard error of the mean of triplicate samples.

**Figure 5 molecules-29-01898-f005:**
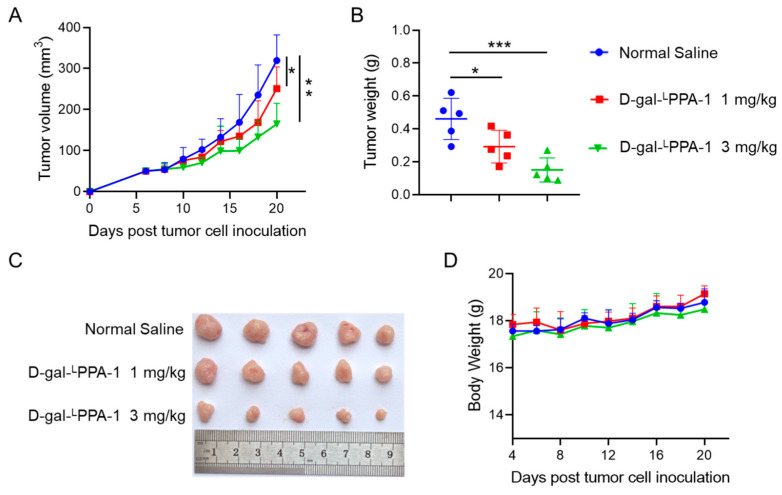
The antitumor effects of the peptide D-gal-^L^PPA-1 on MC38 tumor-bearing mice in vivo. The peptide D-gal-^L^PPA-1 (1 mg/kg), D-gal-^L^PPA-1 (3 mg/kg), or negative control normal saline (NS) was intraperitoneally injected into mice once daily for two weeks. (**A**) The tumor growth curves of MC38 tumors after tumor cell inoculation. (**B**) MC38 tumor weights. (**C**) Photos of tumors removed from MC38 tumor-bearing mice after D-gal-^L^PPA-1 treatment. (**D**) Body weight of the mice. Inter-group differences were statistically analyzed using the unpaired Student’s *t*-test. Data are presented as means ± SEM. Statistical significance was considered when the *p*-value was less than 0.05, denoted as * for *p* < 0.05, ** for *p* < 0.01, and *** for *p* < 0.001.

**Figure 6 molecules-29-01898-f006:**
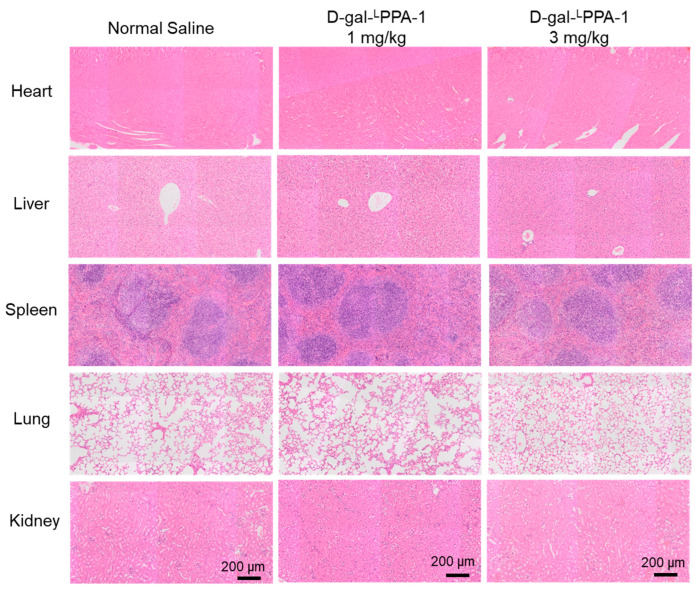
H&E staining of various organs of mice treated with the peptide D-gal-^L^PPA-1 (1 mg/kg), D-gal-^L^PPA-1 (3 mg/kg), or the negative control normal saline. Scale bar: 200 μm.

**Figure 7 molecules-29-01898-f007:**
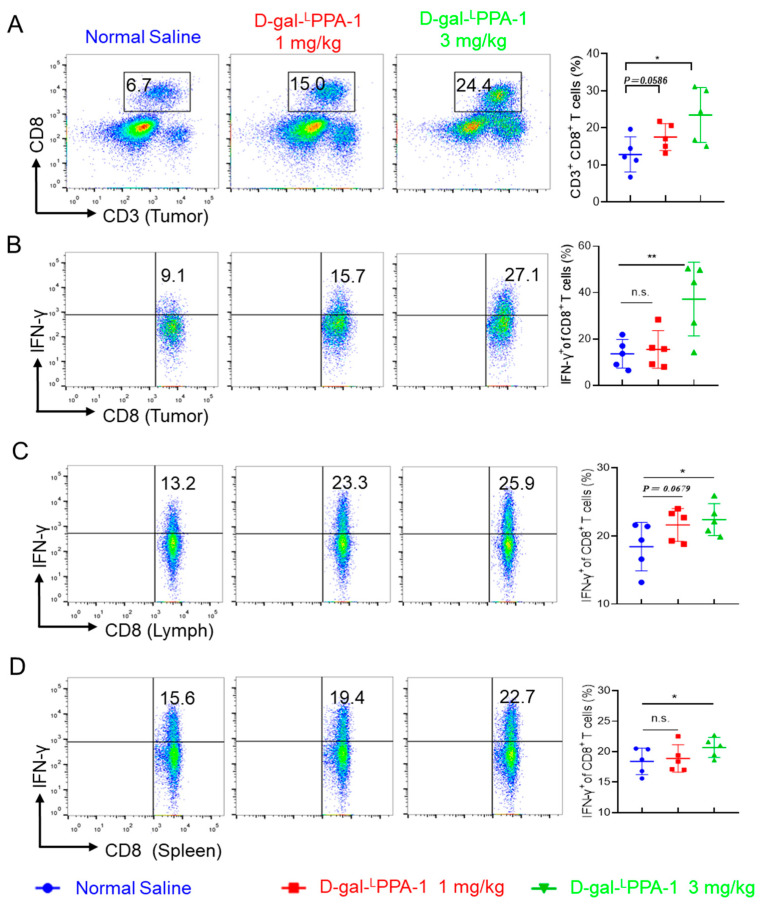
D-gal-^L^PPA-1 significantly enhanced the functionality of CD8^+^ T cells in MC38 tumor-bearing mice. (**A**) The proportion of CD8^+^ T cells that infiltrated the tumor. (**B**–**D**) The proportions of IFNγ^+^CD8^+^ T cells detected by intracellular cytokine staining in cell suspensions from (**B**) tumor tissues, (**C**) draining lymph nodes, and (**D**) spleens. (*n* = 5, mean ± SEM). n.s., not significant; * *p* < 0.05; ** *p* < 0.01.

## Data Availability

Data are contained within the article and Appendix A.

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
