# Peer review of "Development of Glycosylation-Modified DPPA-1 Compounds as Innovative PD-1/PD-L1 Blockers: Design, Synthesis, and Biological Evaluation"

_molecules, 2024, doi:10.3390/molecules29081898_

Round 1
Reviewer 1 Report
Comments and Suggestions for Authors
This manuscript contains the preparation and evaluation of glycopeptides with the ability to bind to PD-1, whose overexpression is related to cancers. The study is thorough and interesting overall. However, the synthetic part is not presented in an analogous fashion and since new molecules are synthesized, more detailed information should be incorporated in the manuscript.
Comments for the authors:
1) The rational behind this work is not quite clear: Specifically, the authors should revise the manuscript explaining the reason why they synthesized the L-aminoacid sequence of the PD-1 receptor peptide ligand and why were these two glycopeptides synthesized as well. Was the reason for glycosylation only to make the L-peptides metabolically more stable or to enhance binding affinity or else?
2) The authors should provide more information on their design: How was the position of the peptide sequence chosen to introduce the carbohydrates? Were there any theoretical or computer-aided studies to indicate that these glycopeptides bind to the same active site in PD-1? If yes, please provide the data.
3) Synthesis: Please provide information on the glycosylation reaction, the purification and yield of the final products. Were the final peptides analysed by HPLC for purity?
4) Please provide information on the cell lines that were chosen: MC38, the Chinese hamster ovary cell line CHO-122 K1-hPD-L1, the CHO-K1-mPD-L1 line, and the human T lymphocytic leukemia. Do all express PD-L1?
5) 2.5. Binding Assay. “The labeled proteins were incubated with a certain concentration of peptide solution at the same volume” Please provide detailed information on the concentrations used.
6) 3.1. Inhibition of PD1-PD-L1 interaction. In this experiment please provide more information on the various concentrations of the peptides that were incubated with the cells. The amount of PD-L1 (50ng) used in the experiment, was it is excess compared to the various dilutions of the glycopeptides?
7) 3.2. I have not understood in Fig. 2B what do the bars (in shades of grey and white) represent? Please explain and revise accordingly.
8) The discussion section is too brief. The authors should discuss appropriately the differences in the biological properties between the two glycopeptides and also, it would be useful to discuss the biological properties of these new glycopeptides in comparison to the D-aminoacid analogues.
Comments on the Quality of English LanguageMinor English editing would be helpful, but no problems identified.
Reviewer 2 Report
Comments and Suggestions for Authors
Comment,
According to the similarity check, authors should cite the references where they token especially in materials and methods.
The statistical analysis is not clear, please specify the model used.
In the discussion section, you should discuss the exhibited robust inhibitory activity against the interaction between PD-1 and PD-L1 as affected by D-gal-LPPA-1.
Overall the discussion section is very poor.
Authors should follow the journal format.
The figures are well-designed.
Comments on the Quality of English Language
Minor corrections
Reviewer 3 Report
Comments and Suggestions for Authors# Title looks little odd: Authors may write Biological Evaluation instead of activity Study.
# Page 3 line 1: Solid-Phase Peptide Synthesis of what? Please insert name or number of the compound.
# Detail synthesis procedure and characterization data should be inserted in the experimental section.
# Discussion section is too low. Should be expanded. Seems results section included discussion.
Reviewer 4 Report
Comments and Suggestions for Authors
The manuscript described the design, synthesis and biological activities of glycosylation-modified D PPA-1 compounds as innovative PD-1/PD-L1 blockers.
There are some questions about the manuscript as follows:
1. Authors do not provide the synthetic procedure, chemical structures, components, characterziation, purity dada of the target compounds.
2. There are some errors in typos, spelling, syntax, punctuation, usage of abbreviation, consistency in language style in the manuscript.
Based on the above, I do not think this manuscript should be accepted for publication in MOLECULES.
Comments on the Quality of English LanguageThe manuscript described the design, synthesis and biological activities of glycosylation-modified D PPA-1 compounds as innovative PD-1/PD-L1 blockers.
There are some questions about the manuscript as follows:
1. Authors do not provide the synthetic procedure, chemical structures, components, characterziation, purity dada of the target compounds.
2. There are some errors in typos, spelling, syntax, punctuation, usage of abbreviation, consistency in language style in the manuscript.
Based on the above, I do not think this manuscript should be accepted for publication in MOLECULES.
Round 2
Reviewer 2 Report
Comments and Suggestions for Authors
Accept
Comments on the Quality of English LanguageMinor editing
Reviewer 3 Report
Comments and Suggestions for Authors
Accept
Reviewer 4 Report
Comments and Suggestions for Authors
The authors have responded the key comments in this revision, I think this manuscript should be accepted for publication in MOLECULES.
Comments on the Quality of English LanguageThe authors have responded the key comments in this revision, I think this manuscript should be accepted for publication in MOLECULES.